# Assessment of the Pharmacokinetics, Disposition, and Duration of Action of the Tumour-Targeting Peptide CEND-1

**DOI:** 10.3390/ijms24065700

**Published:** 2023-03-16

**Authors:** Harri A. Järveläinen, Christian Schmithals, Maike von Harten, Bianca Kakoschky, Thomas J. Vogl, Stephen Harris, Claire Henson, Gemma Bullen-Clerkson, Albrecht Piiper

**Affiliations:** 1Department of Food Hygiene and Environmental Health, Faculty of Veterinary Medicine, University of Helsinki, 00014 Helsinki, Finland; 2Department of Medicine 1, University Hospital Frankfurt, 60590 Frankfurt, Germany; 3Department of Diagnostic and Interventional Radiology, University Hospital Frankfurt, 60590 Frankfurt, Germany; 4Department of Metabolism, Pharmaron UK, Pegasus Way, Crown Business Park, Rushden, Northamptonshire NN10 6ER, UK; 5German Cancer Consortium (DKTK) and German Cancer Research Center (DKFZ), 69120 Heidelberg, Germany

**Keywords:** CEND-1, iRGD peptide, LSTA1, certepetide, pharmacokinetics, quantitative whole-body autoradiography (QWBA), quantitative radioactivity analysis (QRA)

## Abstract

CEND-1 (iRGD) is a bifunctional cyclic peptide that can modulate the solid tumour microenvironment, enhancing the delivery and therapeutic index of co-administered anti-cancer agents. This study explored CEND-1’s pharmacokinetic (PK) properties pre-clinically and clinically, and assessed CEND-1 distribution, tumour selectivity and duration of action in pre-clinical tumour models. Its PK properties were assessed after intravenous infusion of CEND-1 at various doses in animals (mice, rats, dogs and monkeys) and patients with metastatic pancreatic cancer. To assess tissue disposition, [^3^H]-CEND-1 radioligand was administered intravenously to mice bearing orthotopic 4T1 mammary carcinoma, followed by tissue measurement using quantitative whole-body autoradiography or quantitative radioactivity analysis. The duration of the tumour-penetrating effect of CEND-1 was evaluated by assessing tumour accumulation of Evans blue and gadolinium-based contrast agents in hepatocellular carcinoma (HCC) mouse models. The plasma half-life was approximately 25 min in mice and 2 h in patients following intravenous administration of CEND-1. [^3^H]-CEND-1 localised to the tumour and several healthy tissues shortly after administration but was cleared from most healthy tissues by 3 h. Despite the rapid systemic clearance, tumours retained significant [^3^H]-CEND-1 several hours post-administration. In mice with HCC, the tumour penetration activity remained elevated for at least 24 h after the injection of a single dose of CEND-1. These results indicate a favourable in vivo PK profile of CEND-1 and a specific and sustained tumour homing and tumour penetrability. Taken together, these data suggest that even single injections of CEND-1 may elicit long-lasting tumour PK improvements for co-administered anti-cancer agents.

## 1. Introduction

There is a substantial volume of literature reporting that poor penetration of anti-cancer agents into solid tumours is an important factor limiting their efficacy. Adequate drug penetration into the tumour can be compromised by several factors, including the dense connective tissue stroma of the tumour microenvironment [1]. This physical barrier is particularly prominent in stroma-rich tumours such as pancreatic ductal adenocarcinoma [2]. Poor drug penetration likely contributes to therapy resistance and the high mortality rate associated with this cancer [3].

The tumour-penetrating peptide CEND-1 (scientifically also known as iRGD) has been shown to improve the tumour penetration of various simply co-administered chemotherapeutics, leading to an improved drug dose efficacy [1]. Since CEND-1 enhances the penetration of anticancer therapeutics specifically into tumours, but not into normal tissues, it also holds the potential for dose reductions, which can attenuate side effects. CEND-1 is a nine-amino-acid cyclic tumour-penetrating peptide (amino acid sequence: CRGDKGPDC) which acts through a three-step process. First, the arginine-glycine-aspartic acid (RGD)-motif binds to αvβ3 and αvβ5 integrin receptors in the tumour vascular endothelium, triggering protease cleavage of the peptide. The cleaved CEND-1 fragment is then able to bind to neuropilin-1. This causes tumour extravasation of the peptide and of co-administered payloads by a transendothelial endocytic mechanism [1,4,5,6] that resembles vascular endothelial growth factor-A (VEGF-A)-induced transport vesicles called vesiculo-vacuolar organelles [7]. CEND-1/iRGD then activates a transcellular transport pathway in the extravascular tumour tissue, effectively transforming the solid tumour microenvironment into a temporary drug conduit that facilitates the penetration of chemotherapeutics co-administered with CEND-1 into tumours [1,4,8,9,10]. Healthy blood vessels express little to no αvβ3 and αvβ5 integrin receptors, and therefore have limited CEND-1 homing and pharmacological activity [10]. As such, CEND-1 co-administration may increase the specificity and efficacy of many types of anti-cancer treatments, in a wide variety of solid tumours. A recent phase I trial using CEND-1 in combination with chemotherapy in 31 patients with metastatic pancreatic cancer demonstrated an encouraging safety profile and signs of clinical activity [11], with confirmatory randomised trials planned.

This study examined the pharmacokinetic (PK) properties of CEND-1 in mouse, rat, dog and monkey models for translational purposes. The clinical PK properties were investigated in patients with metastatic pancreatic ductal adenocarcinoma before and during combination therapy with nab-paclitaxel and gemcitabine. Moreover, in vivo distribution and tumour selectivity of CEND-1 was assessed in mice bearing orthotopic 4T1 mammary carcinoma. Finally, we examined the duration of the tumour-penetrating effect of CEND-1 in a mouse model of hepatocellular carcinoma (HCC).

## 2. Results

### 2.1. In Vivo Pharmacokinetic Profile in Mice, Rats, Dogs, and Monkeys

The in vivo pharmacokinetic assessment of CEND-1 in mice, rats, dogs and monkeys after a single intravenous dose showed pharmacokinetic profiles comparable in all species (Table 1). The systemic exposure of CEND-1, expressed as C_max_ and AUC_last_, generally increased with dose in a more-than-dose-proportional manner, with rapid elimination. The average half-lives were approximately 20–25 min in mice, 30 min in rats and 40 min in dogs. 

Following an IV dose of CEND-1 at 5 and 50 mg/kg in male cynomolgus monkeys, the mean drug exposures (AUC_0-inf_) of CEND-1 at 5 mg/kg and 50 mg/kg were 28,230 hr*ng/mL and 421,119 hr*ng/mL, respectively (Table 1). The increase in drug exposure was higher than dose-proportional. The mean values of C_0_ were 55,084 and 602,161 in 5 mg/kg and 50 mg/kg dose groups, respectively. In monkeys, the average half-life was approximately 55 min.

Detailed clinical observations and cage-side observations indicated no test-article-related abnormalities in any of the pre-clinical species tested. 

### 2.2. Clinical Pharmacokinetics

Human PK parameters following CEND-1 monotherapy (run-in day 1) and combination therapy with nab-paclitaxel and gemcitabine (cycle 1 day 1 and cycle 6 day 1) at 1.6 and 3.2 mg/kg CEND-1 are summarised in Figure 1. Furthermore, detailed PK parameters at 3.2 mg/kg CEND-1 are shown in Table 2. Following the administration of CEND-1 at 1.6 and 3.2 mg/kg at run-in day 1, plasma concentrations increased rapidly with peak plasma concentration occurring at the first post-dose PK sampling timepoint (3 min post-dose), before decreasing in an exponential manner (Figure 1A). Mean concentrations had fallen to 439 ng/mL (SD = 317.9) by 8 h post-dose. 

Administration of subsequent CEND-1 doses at Cycle 1 Day 1 and Cycle 6 Day 1 again resulted in plasma concentrations rapidly increasing with peak concentrations occurring at the first post-dose PK sampling timepoint, before decreasing in an exponential manner (Figure 1B,C). Mean plasma concentrations at each CEND-1 dose level declined in a similar manner in the 8 h post-dosing.

Assessment of plasma CEND-1 parameters demonstrated that exposure (AUC_0-t,_ AUC_0-6h_ and AUC_0-inf_) followed the same pattern described for C_max_ with a linear increase with an increased dose. The median half-life values of CEND-1 (T_1/2_) were between 1.6 and 1.8 h over PK sampling timepoints. Mean clearance (CL) values were between 106.8 mL/h/kg and 266.5 mL/h/kg. The terminal volume of distribution (Vz) mean values were between 220.9 mL/kg and 277.4 mL/kg over all days of PK sampling.

The PK profile of CEND-1 at the beginning of the study (run-in and C1D1) and after five completed cycles of therapy (C6D1) were similar.

### 2.3. Tumour-Specific Distribution of [^3^H]-CEND-1 Assessed by QWBA and QRA

The quantitative whole-body tissue distribution (QWBA) of [^3^H]-CEND-1 in female BALB/c mice with 4T1 tumours was assessed at 30 min and 3 h after intravenous administration (Figure 2; representative examples). At 30 min post-dose, [^3^H]-CEND-1 derived radioactivity was detected in 35 of the 40 tissues assessed, including the tumour (Table 3). Radioactivity at 3 h post-dose was below the limit of detection in 32 of the 40 tissues and was only present in quantifiable amounts in the alimentary canal (especially caecum), kidneys, bladder, eyes and tumour. The tissue with the highest radioactivity at 3 h was the kidney (Table 3). 

The quantitative tissue distribution (QRA) of [^3^H]-CEND-1 in female BALB/c mice with 4T1 tumours was assessed at 30 min, 3 h and 8 h after intravenous administration (Figure 3). At 30 min post-dose, [^3^H]-CEND-1 concentration was highest in tumours, followed by the lungs, liver and spleen. [^3^H]-CEND-1 concentration decreased over 3 h in all tissues, however concentrations remained highest in tumours. By 8 h, [^3^H]-CEND-1 tumour concentration had decreased but were still 2.7–4.1 times higher than in the other organs analysed (0.062 ± 0.044 tumour; 0.023 ± 0.004 spleen; 0.020 ± 0.006 lungs; and 0.015 ± 0.002 liver) (Figure 3). 

### 2.4. Evans Blue Injection to Assess the Durability of the Tumour-Penetrating Effect of CEND-1

The effect and duration of CEND-1 on tumour extravasation and tissue penetration was measured by intravenous injection of albumin-binding Evans blue dye and determination of the dye content of the tissues. In *transforming growth factor-α* (*TGF*α)/c-*myc* mice with radiologically proven HCCs, intravenously injected CEND-1 increases the levels of co-administered Evans blue approximately threefold [12]. Here, we examined the effect of CEND-1 on the tumour levels of Evans blue when the dye was injected 24 h after injection of CEND-1. Photometric quantification of the dye extracted from the tumours revealed an approximately 2.7-fold increase in Evans blue in the tumour tissue from CEND-1-injected mice, whereas the RGD control peptide or saline had no effect (Figure 4). 

We next examined whether the duration of the tumour-penetrating effect of CEND-1 in HCC mice could be monitored in vivo by the clinically translatable Gd-DTPA-enhanced MRI [13]. *TGF*α/c-*myc* mice with HCC according to Gd-EOB-DTPA-enhanced MRI, were subjected to a Gd-DTPA-enhanced MRI performed before and shortly after the injection of CEND-1, RGD control peptide or saline. In the animals pre-treated with CEND-1, Gd-DTPA-injected animals showed a stronger increase in MRI signal intensity (higher % signal increase) in the tumours as compared to the animals injected with RGD control peptide or saline prior to the application of the contrast agent (Figure 5A,B). 

In the next set of experiments, we examined the effect of CEND-1 on tumour enrichment of MRI contrast agent, when the contrast agent was administered either directly after or two days after the injection of CEND-1 (Figure 5C). Surprisingly, the increased accumulation of the contrast agent in the tumours was similar in animals receiving contrast agent either directly after the injection of the peptide or two days later (increasing by 25% vs. 22.5%, Figure 5C). CEND-1 also elicited an increased accumulation of contrast agent in the tumours of nude mice with HepG2 tumours when CEND-1 had been injected 24 h prior (Gd-DOTA caused an increase in the signal intensity of the tumours by 16% directly after injection of CEND-1 and still 11% after 24 h after injection of CEND-1, Figure 6).

## 3. Discussion

CEND-1 has been extensively studied as a potentially broadly applicable therapeutic enhancer of anti-cancer therapeutics [14]. The recently published results from the first-in-man Phase I study showed favourable results in pancreatic cancer patients [11,15]. The present study assessed the CEND-1 PK profile pre-clinically, as well as clinically in patients with metastatic pancreatic ductal adenocarcinoma. In mouse tumour model studies, CEND-1/iRGD was commonly administered at a dose level of 4–5 mg/kg. The pre-clinical PK assessment in this study demonstrated that this dose results in a Cmax (C_0_) of approximately 40 μg/mL. Consequently, this was the set-as-target exposure for the Phase 1 study. Simple allometric scaling indicated that this may be achieved with dose levels of 1.6–3.2 mg/kg in humans.

As shown in this study, the allometric scaling predicted the target exposures in humans well and the clinical PK profile was largely consistent with the pre-clinical species. After intravenous bolus administration, the PK of CEND-1 was monophasic and systemic exposures generally increased with dose in a more-than-dose-proportional manner, possibly due to a slightly faster metabolism at lower doses. The PK was characterised by a rapid distribution phase and linear elimination phase with a half-life of approximately 2 h in humans. Although both dose levels in expansion cohorts (1.6 and 3.2 mg/kg) led to an exposure in the predicted active dose range, the 3.2 mg/kg dose resulted in pharmacokinetic characteristics that most closely resembled the most efficacious exposure profile in mouse tumour models. Therefore, 3.2 mg/kg was chosen as the recommended phase 2 dose (RP2D) to be evaluated in further clinical studies. Interestingly, the CEND-1 monotherapy (run-in) PK profile was comparable with the PK profiles following combination chemotherapy, demonstrating that CEND-1 pharmacokinetics are not affected by chemotherapy treatment (both initially and after repeat treatments), suggesting a lack of anti-drug antibody responses. 

As there is no clear relationship between drug concentrations in the plasma and those in tissue [16], and only the target tissue (tumour) concentrations are relevant for efficacy, the whole-body distribution and tumour selectivity of radioactivity following [^3^H]-CEND-1 administration in mice bearing 4T1 mammary carcinoma tumours were assessed using QWBA and QRA. Initially, [^3^H]-CEND-1-related radioactivity was detectable in most organs at 30 min post-administration but was subsequently largely cleared in non-tumour tissues within 3 h post-administration, with the exception of the tumour, and tissues known to be related to the metabolism and secretion of CEND-1 (i.e., the kidneys and bladder). The renal excretion is similar to other tumour-targeting RGD peptides such as cilengitide [17]. Radioactivity was also observed in the alimentary tract, but whether this is related to the metabolism and/or excretion of CEND-1 in the gastrointestinal tract remains to be investigated. 

The pronounced tumour specificity of [^3^H]-CEND-1 became apparent by 3 h post-administration. Although at 3 h the [^3^H]-CEND-1-related signal was mostly eliminated systemically from tumours and healthy tissues, the signal was still detectable in tumours, demonstrating specific, potent and prolonged retention in tumours. Furthermore, the tumour concentrations of radioactivity remained at least seven-fold higher than any other tissue at the 12 mg/kg dose. These results are supportive of previous non-clinical studies that demonstrated the ability of CEND-1/iRGD in enhancing drug delivery and efficacy in both primary and metastatic tumours in a variety of mouse and human tumour types [1]. It is nevertheless possible that the radiolabelled CEND-1 might have shown a slightly reduced deep tumour penetration due to its modification.

Notably, [^3^H]-CEND-1-related radioactivity was not detected in the tissues of the central nervous system including the brain, suggesting that CEND-1 does not cross the blood-brain barrier. Nevertheless, since CEND-1/iRGD has shown efficacy in multiple pre-clinical models of glioblastoma, there is a possibility that the disrupted blood-brain barrier (BBB) may allow peptides such as CEND-1 to penetrate the brain in certain pathological conditions [18].

The target tissue pharmacokinetics may not always correlate with the pharmacodynamic effect of a drug. Although the circulation half-life in mice is only 20 min, alterations in the pancreatic ductal adenocarcinoma tissue were still evident by electron microscopy 24 h after a single dose of CEND-1/iRGD in mice [6]. In that study, silica nanoparticles co-injected with CEND-1 also remained accumulated in the tumour vascular endothelium and tumours 24 h after their co-administration, suggesting that the CEND-1-induced transport system in the tumour, i. e. the pharmacodynamic action of CEND-1, may last much longer than the presence of the drug alone would suggest. The present study shows that the tumour-penetrating effect of CEND-1/iRGD was still present at 24–48 h after injection of the peptide in models of liver cancer, providing further support into the long-lasting tumour-penetrating effect of CEND-1. The longer duration of the tumour-penetrating effect of CEND-1 is in agreement with the ability of the peptide to improve the efficacy of co-injected drugs with long circulation half-lives, such as Trastuzumab [5]. If this applies to tumours in patients, a single injection of CEND-1 may lead to a systemically administered drug having long-lasting access to tumours. The mechanism of the prolonged tumour-penetrating effect of CEND-1 is currently unclear, but it may be a combination of both the prolonged intra-tumour pharmacokinetics demonstrated in this study, as well as a pharmacodynamic action that is not dependent on the continuing presence of CEND-1.

The tumour-penetrating effect of CEND-1 may vary depending on the tumour entities and tumour model. Data from an earlier study utilising the 4T1 breast cancer xenograft mouse model suggested the penetration effect to only last approximately one hour with this experimental approach [19]. This study also found a correlation between pharmacokinetic properties of the different peptide derivates and the duration of activity.

In summary, these results indicate a favourable in vivo pharmacokinetic profile of CEND-1 after intravenous administration and demonstrate effective and long-lasting tumour-homing properties (i.e., greater tumour selectivity and prolonged binding relative to health tissues). The initial clinical trial of CEND-1 in a phase I trial demonstrated an acceptable safety profile of CEND-1 as a monotherapy and in combination with chemotherapy drugs [11]. The present study’s findings, demonstrating favourable pharmacokinetics, tumour-specific accumulation and the prolonged efficacy of CEND-1, coupled with the recent Phase I study findings, support the suitability for further human investigations of CEND-1 in the treatment of cancer. 

## 4. Materials and Methods

The cyclic iRGD/CEND-1 peptide [sequence: CRGDKGPDC] and RGD control peptide (CRGDDGPKC) for pre-clinical use was sourced either from GenScript (Piscataway, NJ, USA) or CPC Scientific (Hangzhou, China). It was manufactured using solid phase peptide synthetic techniques with high chemical purity. It has a cyclic structure (S-S bonded through the cysteine side chains). The purity of all materials was >95%.

The CEND-1 drug product for clinical use was provided by Cend Therapeutics, Inc. (Basking Ridge, NJ, USA). CEND-1 for injection was a sterile, white, lyophilised powder supplied as 100 mg per vial of active ingredient dose strength for intravenous administration. CEND-1 injection consisted of the CEND-1 drug substance with sodium acetate trihydrate and mannitol as excipients.

The animal pharmacokinetic studies were conducted under approvals from Pharmaron Institutional Animal Care and Use Committee. The murine orthotopic 4T1 tumour model animal experimentation was conducted under approvals from the Animal Welfare and Ethical Review Body (AWERB) (License number PEDE7C911, Protocol 8; study numbers CDX/01, CDX/02, CDX/03) and in accordance with the Animal Welfare Act 2006, with UK Home Office Guidance on the implementation of the Act and with all applicable Codes of Practice for the care and housing of laboratory animals. Experiments involving the *TGFα/c-myc* bitransgenic mice and tumour-induced NMRI Foxn1 nude mouse models were approved by the local animal care committee in agreement with German legal requirements (approval number FK1100 RP Hessen). The clinical pharmacokinetic study protocol and all amendments were approved by Central Adelaide Local Health Network Human Research Ethics Committee (HREC) (HREC/18/CALHN/225), St John of God Hospital Ethics Office (Ref 1397) and Sydney Local Health District HREC, Concord Repatriation General Hospital on behalf of Alfred Hospital (HREC/18/CRGH/72). All human participants provided written informed consent prior to enrolment. The human study is registered with ClinicalTrials.gov (NCT03517176) and the Australian New Zealand Clinical Trials Registry (ACTRN12618000804280).

### 4.1. Pre-Clinical Pharmacokinetics

The in vivo pharmacokinetic profile of CEND-1 after a single dose was evaluated in four non-GLP animal studies including Sprague-Dawley rats, Beagle dogs, cynomolgus monkeys and BALB/c mice following the intravenous injection of CEND-1 at various doses (1, 5 and 50 mg/kg for rats, dogs and monkeys; 1.5, 4.5 and 13.5 mg/kg for mice) (Table 1). Blood samples were collected for pharmacokinetic evaluations pre-dose and at 1, 5, 10, 15 and 30 min, and 1, 2 and 6 h post-dose in rats, dogs and monkeys, and 3, 10, 30 and 90 min, and 4 and 8 h post-dose in mice. 

Plasma CEND-1 was quantified by validated liquid chromatography coupled with a tandem mass spectrometry (LC/MS/MS) assay. Tolbutamide or leuprorelin acetate were used as internal standards. AB Sciex Triple Quad API 5500 (Sciex, Framingham, MA, USA), operated in the positive electrospray ionization (ESI) mode, was used to monitor the precursor to production ion transitions of m/z 495.4→69.9.

### 4.2. Design of the Clinical Pharmacokinetics Study

In this study, CEND-1 PK data from up to 28 participants receiving either 1.6 mg/kg (*n* = 14) and 3.2 mg/kg (*n* = 14) in the expansion cohort were analysed. Details of this phase I clinical study, conducted in patients with metastatic pancreatic ductal adenocarcinoma receiving CEND-1 in combination with nab-paclitaxel and gemcitabine, are described in detail by Dean et al. [11]. PK parameters were measured following CEND-1 monotherapy (run-in day 1) and combination therapy (cycle 1 day 1 and cycle 6 day 1, if applicable). 

In the initial monotherapy run-in phase, single doses of CEND-1 were given one to seven days before the start of the combination chemotherapy. In the combination therapy phase, patients first received an intravenous infusion of nab-paclitaxel (125 mg/m^2^) over 30 min (±3 min). CEND-1 was then given intravenously as a slow IV push over 1 min (±30 s). The intravenous infusion of gemcitabine (1000 mg/m^2^ over 30 min (±3 min)) was started within 10 min of CEND-1 administration. On all occasions, plasma samples were collected from patients before the CEND-1 infusion and at 3 min (±1 min), 15 min (±3 min), 30 min (±3 min), 1 hour (±5 min), ¾ h (±10 min) and 6/8 h (±10 min) after completion of the infusion.

The study was conducted at three centres in Australia: The Queen Elizabeth Hospital, Woodville South, South Australia; St John of God Hospital, Subiaco, Western Australia; and The Alfred Hospital, Melbourne Victoria between 13 August 2018 and 19 June 2020. 

### 4.3. [^3^H] CEND-1 Radiolabellin

CEND-1 (Figure 7A), manufactured according to Good Manufacturing Practices, was provided by Cend Therapeutics, Inc. CEND-1 was radiolabelled with N-succinimidyl [2,3-^3^H] propionate ([^3^H]-NSP) (Figure 7B). Briefly, 1.6 mCi aliquots of [^3^H]-NSP were dispensed, and the solvent was evaporated under a gentle stream of nitrogen gas at ambient temperature. CEND-1 peptide in PBS (0.5 mg/mL) was added to the [^3^H]-NSP and re-dissolved in PBS pH 7.4, followed by gentle mixing for approximately 2 h, or overnight at 2–8 °C, until protein tagging and hydrolysis of unbound NSP was complete. The NSP-peptide solution was then purified by reverse phase chromatography and the resulting solution was concentrated using a centrifugal evaporator. The resulting material was dissolved in 1:1 ethanol:water and stored at –80 °C prior to formulation for dose preparation. Radiochemical purity and specific activity of the [^3^H]-labelled test peptide was determined using reverse phase HPLC and quantitative radioactivity analysis (QRA). However, the pharmacologic activity of the labelled peptide was not determined. 

### 4.4. Murine Orthotopic 4T1 Tumour Model 

The 4T1 triple negative breast tumour cell line was obtained from Crown Bioscience UK (Belton, Loughborough, UK). To produce 4T1 tumours, female BALB/c nude mice (8–12 weeks, 20–25 g), obtained from Charles River Ltd. (Margate, Kent, UK), were orthotopically injected in the mammary fat pad with 2 × 10^7^ cells suspended in PBS. 

### 4.5. Quantitative Whole Body Autoradiography (QWBA)

Three female BALB/c mice were injected with 1 × 10^6^ 4T1 cells into the mammary fat pads to induce tumour development. [^3^H]-CEND-1 was administered when the tumours were approximately 2 weeks old (approximately 80–250 mm^3^). Radiation was quantified in tissues using QWBA (representative image is shown in Figure 2). Blood samples were collected at 30 min, 1 and 3 h post-CEND-1 administration. Immediately following blood collection, the mice were perfused using ~20 mL of PBS containing 1% BSA via a 25G 5/8 fixed needle into the left ventricle and making a cut to the right atrium. Attention was paid to minimise contamination of the mouse by the perfusate. Immediately following perfusion, each carcass was snap frozen by immersion in a hexane/dry ice mixture immediately and then stored at −20 °C until QWBA analysis. 

### 4.6. Quantitative Radioactivity Analysis (QRA) 

Female 4T1 tumour-bearing BALB/c mice (as above) each received a single intravenous administration of [^3^H]-CEND 1 at either 1.5, 5 or 12 mg/kg (*n* = 5 mice per dose). The mice were deeply anesthetised to a surgical plane with isoflurane and a blood sample collected by cardiac puncture at 0.5-, 3- or 8-h post-dose administration. The mice were then perfused by using ~20 mL of PBS containing 1% BSA via a 25G 5/8 fixed needle into the left ventricle and making a cut to the right atrium. Immediately following perfusion, the mice were killed by cervical dislocation and the tissues were dissected for QRA analysis. 

### 4.7. Murine TGFa/c-myc HCC Model 

Homozygous metallothionein/*TGFα* and albumin/*c-myc* single transgenic mice in CD13B6CBA background were crossed to generate the male *TGFα/c-myc* bitransgenic mice, as previously described [20,21]. Hepatocarcinogenesis was accelerated by giving the mice ZnCl_2_ (via the drinking water) to induce the expression of *TGFα*. The animals were inspected every 2 to 3 days. To detect and monitor the endogenously formed HCCs in the *TGFα/c-myc* mice, gadoxetic acid (Gd-EOB-DTPA)-enhanced MRI was performed on a 3T MRI scanner (Siemens Magnetom Trio, Siemens Medical Solutions) as described recently [12,21,22]. Mice with HCC as identified by Gd-EOB-DTPA-enhanced MRI were used in the subsequent experiments.

### 4.8. Human HCC Xenograft Model 

HepG2 cells were obtained from American Type Culture Collection (ATCC) and were grown in DMEM supplemented with 10% FBS and penicillin/streptomycin (Life Technologies; ref. [23]). The cells were authenticated by SNP genotyping within the past 6 months. The cells were regularly monitored for morphologic and growth characteristics and mycoplasma contamination. A total of 5 × 106 cells (resuspended in 100 µL of PBS) were injected subcutaneously into the flanks of NMRI Foxn1 nude mice (Harlan Laboratories B.V., Indianapolis, IN, USA). The mice were assigned to the treatment groups 3 to 4 weeks post-inoculation with the tumour cells. For the tumour penetration experiments, the mice were used 5 to 7 weeks post-inoculation.

### 4.9. Tumour Penetration Assessment with Evans Blue

TGFα/c-myc mice with confirmed liver tumours according to Gd-EOB-DTPA (Dotarem)-enhanced MRI or nude mice with HepG2 tumours received iRGD/CEND-1 or RGD control peptide (4 μmol/kg each), or PBS by tail vein injection. Evans blue (33.3 mg/kg, MP Biomedicals, Eschwege, Germany) was injected intravenously 15 min later. Another 30 min later, the mice were terminally perfused with Ringer solution by cannulation of the left heart ventricle. Organs were inspected macroscopically following laparotomy, and the liver, tumour and other organs were excised. Evans blue was extracted from tissues in N,N-dimethylformamide for 24 h at 37 °C and quantified spectrophotometrically (Beckman Coulter DU 800, Beckman Coulter, Brea, CA, USA) at 600 nm. 

### 4.10. Tumour Penetration Assessment with Gd-DTPA–Enhanced MRI 

The *TGFα/c-myc* mice with confirmed liver tumours or nude mice with HepG2 xenografts were anaesthetised by intraperitoneal injection of ketamine (70 mg/kg body weight) and xylazine (10 mg/kg body weight), followed by a basal T1–weighted MRI and immediately thereafter a Gd-DTPA-enhanced MRI [21]. In the nude mice experiments, Gd-DOTA was used instead of Gd-DTPA. Either CEND-1 or RGD control peptide (4 µmol/kg each via tail vein) was injected 12–24 h later, followed by a basal and a Gd-DTPA-enhanced MRI [13]. For quantitative analyses of the MRI data, signal intensities were measured with operator-defined regions of interest (ROI) drawn into the images as described recently [22]. ROIs were placed in the livers and tumours. Signal intensity changes were calculated by subtracting the pre-contrast values from the signal intensities obtained upon addition of Gd-DTPA. The alterations of the signal intensity in the tumours and livers due to CEND-1 or the RGD control peptide were expressed as a fold increase in the signal intensity by Gd-DTPA preinjected with PBS.

### 4.11. Statistical Analysis

No formal sample size calculation was performed. Samples of 2–12 animals and 31 pancreatic cancer patients were considered acceptable to perform initial pharmacokinetic evaluations. The concentrations of CEND-1 were used to calculate PK parameters by employing a non-compartmental analysis (Phoenix^TM^ WinNonlin, version 6.1). The linear log trapezoidal algorithm, with a weighting of 1/Y*Y was used for parameter calculations. Mean PK parameters were calculated from individual animals in each treatment group. Concentrations below the LLOQ (if any) were excluded for the calculation of PK parameters. The mean values of PK parameters were calculated using Microsoft Excel version 2010. 

## Figures and Tables

**Figure 1 ijms-24-05700-f001:**
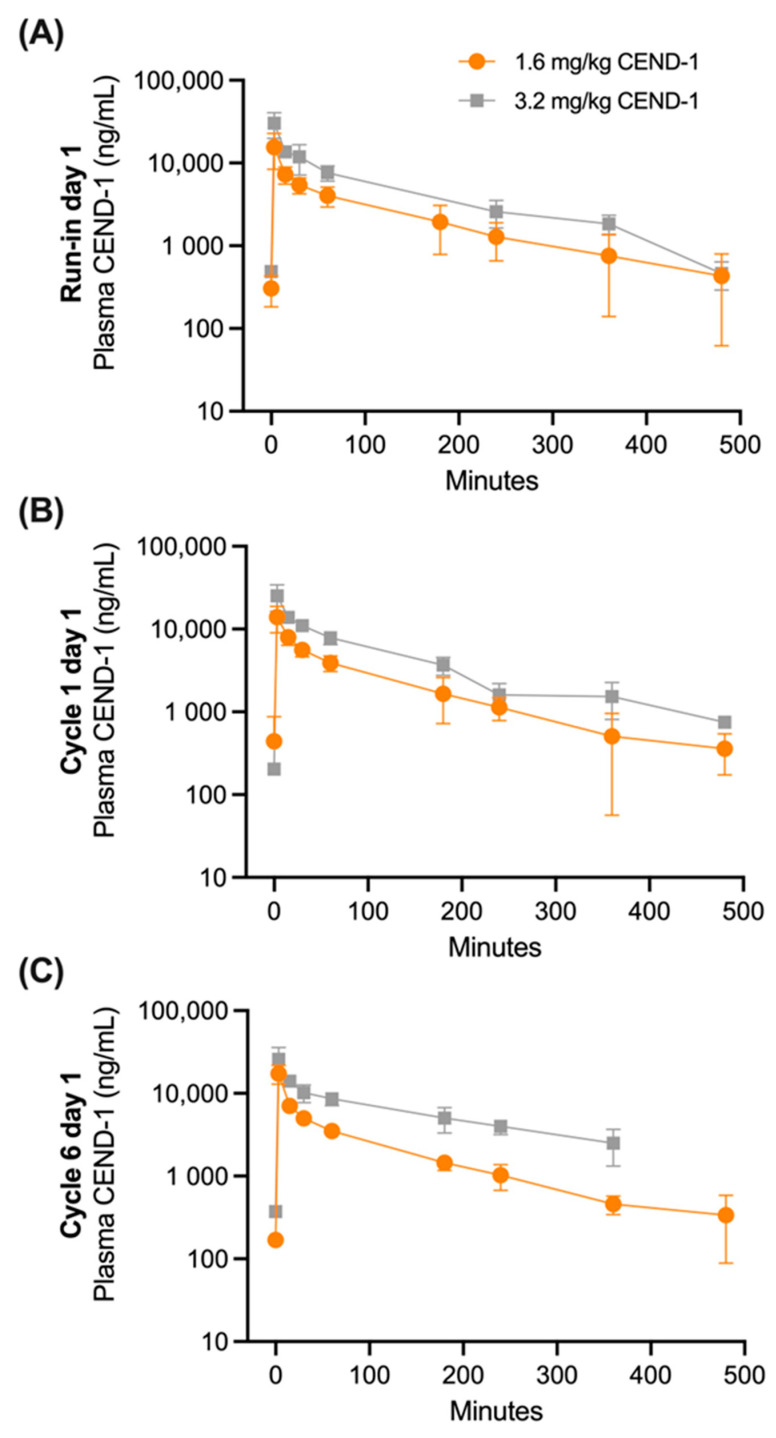
Mean (±SD) plasma CEND-1 concentrations over time in pancreatic cancer patients by dose level (1.6 mg/kg or 3.2 mg/kg) at (**A**) run-in prior to chemotherapy treatment (*n* = 7–14 patients), (**B**) cycle 1 day 1 of nab-paclitaxel+gemcitabine chemotherapy treatment (N = 9–14) and (**C**) cycle 6 day 1 of nab-paclitaxel+gemcitabine chemotherapy treatment (*n* = 1–8).

**Figure 2 ijms-24-05700-f002:**
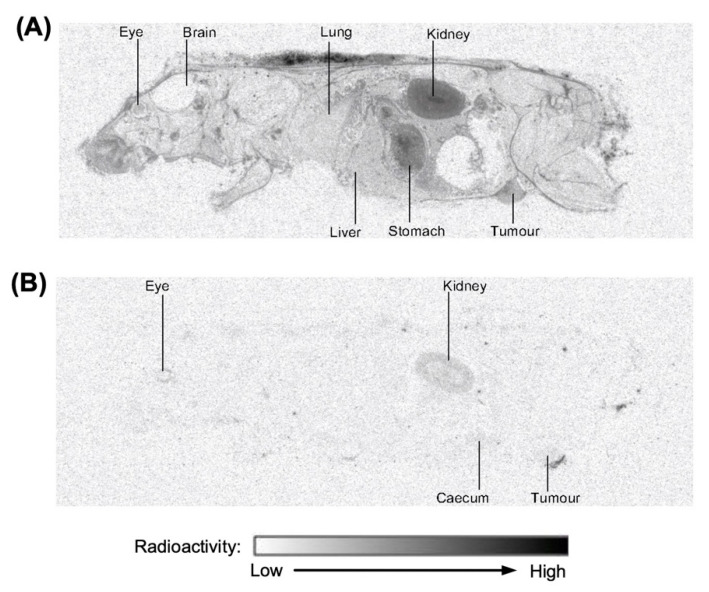
Representative images of distribution of radioactivity in perfused female BALB/c mice with 4T1 cell tumours 0.5 h (**A**) and 3 h (**B**) after intravenous administration of [^3^H]-CEND-1 at a target dose of 5 mg/kg.

**Figure 3 ijms-24-05700-f003:**
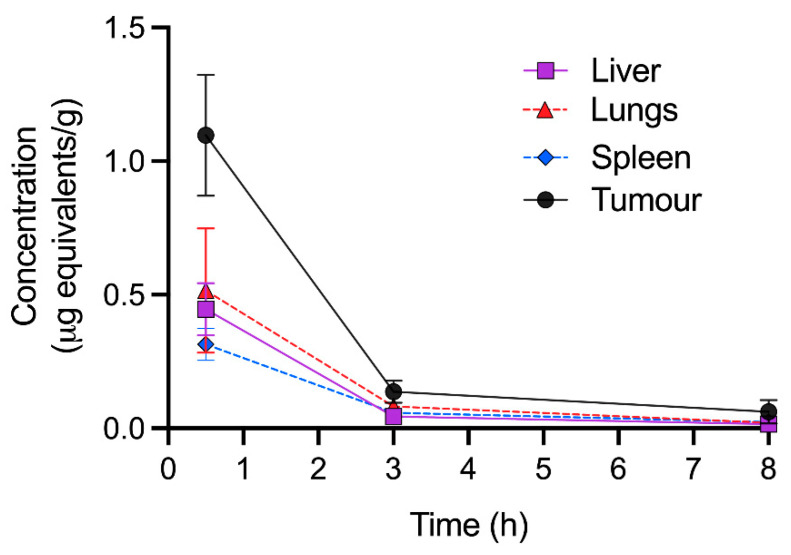
Quantitative radioactivity analysis of perfused tissues at 0.5–8 h following intravenous administration of 5 mg/kg of [^3^H]-CEND-1 in female BALB/c mice with 4T1 cell tumours. The data show the mean ± SD of 5 mice per timepoint.

**Figure 4 ijms-24-05700-f004:**
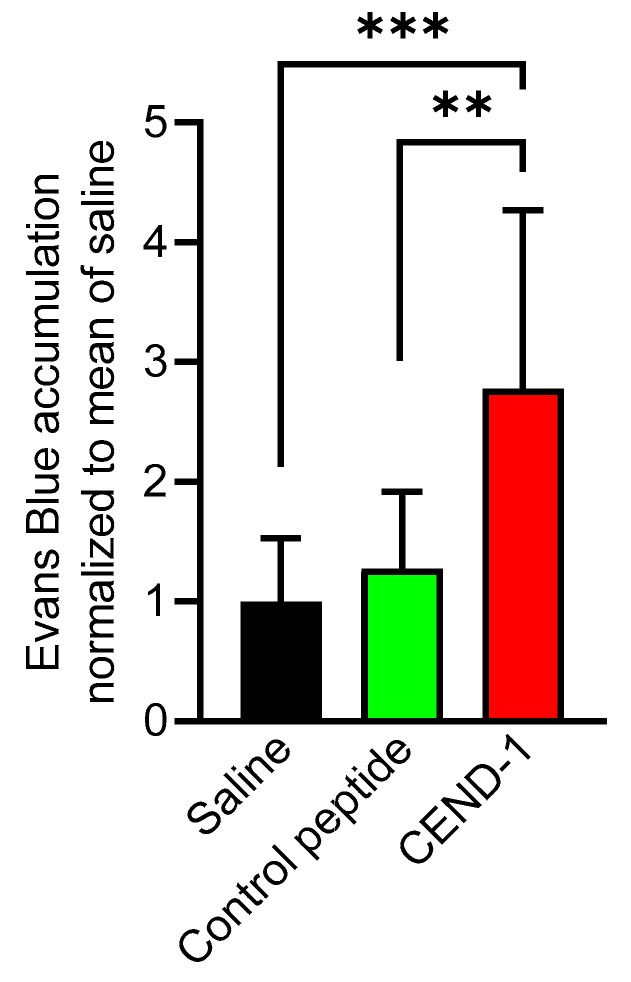
CEND-1 increased the tumour levels of Evans blue in HCCs, when the dye was injected 24 h after injection of the peptide. *TGF*α/c-*myc* mice bearing MRI-verified liver tumours were intravenously injected with 4 μmol/kg of CEND-1 or control peptide (dissolved in PBS), or PBS alone. Then, 24 h later 33 mg/kg bodyweight of Evans blue was i.v. injected into the mice. Evans blue accumulation in the tumours of the mice was quantified. Bars represent mean ± SD; ** *p* = 0.01; *** *p* < 0.001 (one-way ANOVA with Tukey’s multiple comparisons test); *n* = 11 tumours from four animals.

**Figure 5 ijms-24-05700-f005:**
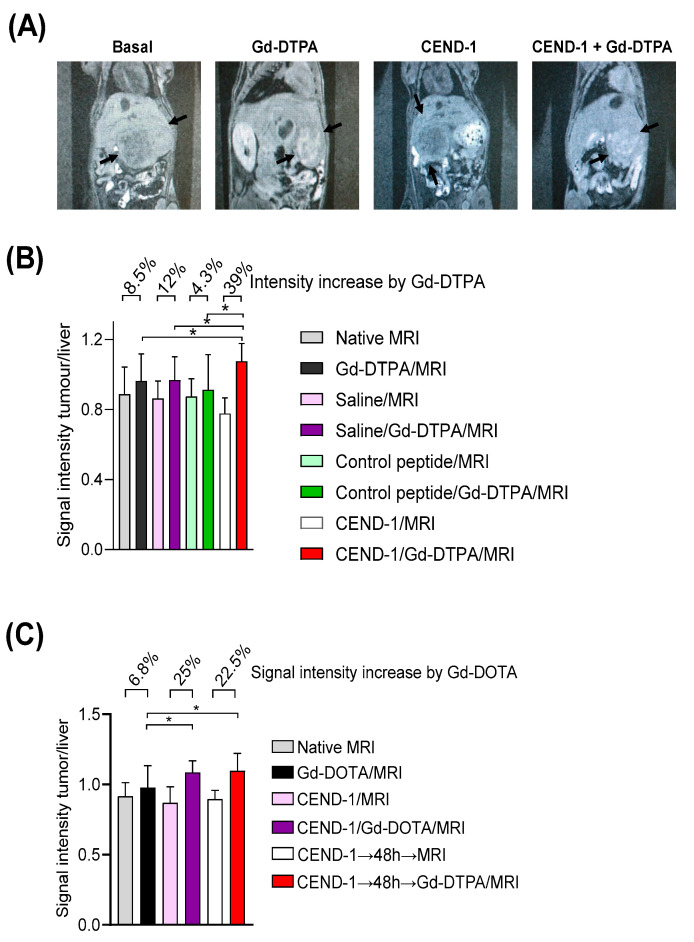
Injection of CEND-1 into *TGF*α/c-*myc* mice with HCC caused an increased accumulation of contrast agent in the tumour, when the contrast agent was administered 48 h later. (**A**) CEND-1 increased the MRI signal in the tumour of *TGFα*/*c-myc* HCC mice. Arrays indicate the tumours. (**B**) Ratio of tumour/liver signal intensity following the treatments. Asterisks indicate a significant difference * *p* < 0.05 compared to the three previous measurements. N = 8 per group. (**C**). CEND-1 elicited a prolonged increase in tumour penetrability in *TGFα/c-myc* HCC mice. *TGFα/c-myc* mice with suitable tumours according to Gd-EOB-DTPA-enhanced MRI received MRI before and after the injection of Gd-DTPA. After 2 days, MRI was performed before and after the injection of CEND-1, followed by the injection of Gd-DTPA and another MRI. Another 2 days later, the mice received additional Gd-DTPA-enhanced MRI. Bars represent mean ± SD; asterisks indicate significant differences (One way ANOVA with Tukey’s multiple comparisons test). * *p* < 0.05. *n* = 11–16.

**Figure 6 ijms-24-05700-f006:**
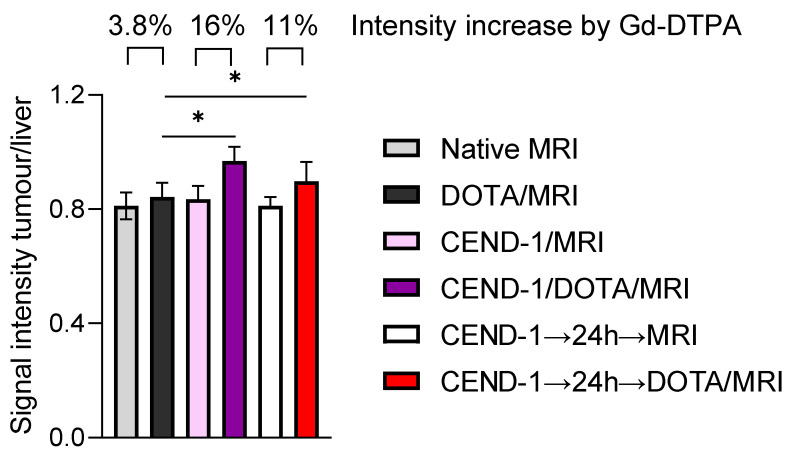
CEND-1 elicited a prolonged increase in tumour penetrability in HepG2 HCC mice as detected by the effect of CEND-1 on contrast agent accumulation in the HCCs. Nude mice with HepG2 tumours received DOTA-enhanced MRI before and after the injection of CEND-1. After 24 h, another DOTA-enhanced MRI was performed. Bars represent mean ± SD; asterisks indicate significant differences (paired T-test). * *p* < 0.05. *n* = 8 per group.

**Figure 7 ijms-24-05700-f007:**
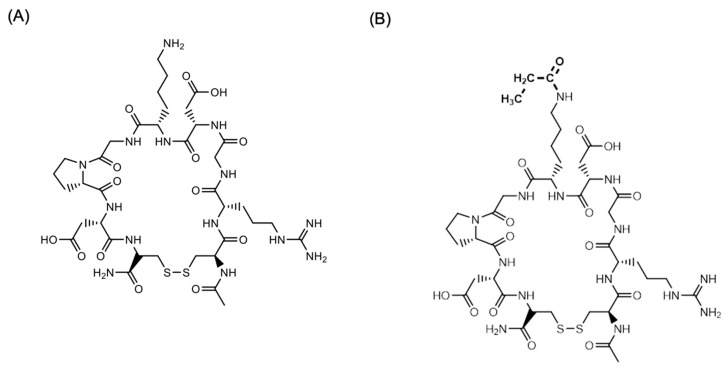
Structure of (**A**) CEND-1 and (**B**) [^3^H]-CEND-1. In (**B**) the H atoms of the acetyl group were the ^3^H.

**Table 1 ijms-24-05700-t001:** Derived mean pharmacokinetic parameters for CEND-1 in mouse, rat, dog and monkey plasma after a single intravenous infusion of CEND-1 (±SD).

Number of Animals (Sex)	CEND-1 Dose (mg/kg)	t_1/2_ h	C_0_ (ng/mL)	V (mL/kg)	Cl_obs (mL/hr/kg)	AUC_0-inf_ (h*ng/mL)
**Mouse ***						
3 (M)	1.5	0.306	10,343	449	1016	1476
3 (M)	4.5	0.344	40,291	599	1205	3695
3 (M)	13.5	0.547	68,358	1007	1277	10,569
**Rat ***						
6 (M)	1	0.805	4230	640	558	1770
6 (F)	1	0.248	3983	201	562	1751
6 (M)	5	0.460	24,333	366	552	9058
6 (F)	5	0.437	28,400	323	513	9747
6 (M)	75	0.341	469,000	171	348	215,476
6 (F)	75	0.391	436,333	254	451	166,331
**Dog**						
3 (M)	1	0.665 (0.0448)	6010 (1985)	241 (27)	253 (40.1)	4030 (686)
3 (F)	1	0.668 (0.0312)	4777 (496)	253 (25.4)	255 (20.9)	3946 (328)
3 (M)	5	0.655 (0.0506)	30,200 (656)	230 (23.1)	243 (8.87)	20,617 (765)
3 (F)	5	0.648 (0.0486)	25,133 (3635)	230 (14.6)	247 (32.9)	20,443 (2530)
5 (M)	75	0.615 (0.0361)	461,000 (61,745)	204 (17.6)	230 (10.0)	326,748 (14,280)
5 (F)	75	0.620 (0.0439)	475,800 (43,900)	208 (16.5)	233 (18.3)	323,936 (25,283)
**Monkey**						
3 (M)	5	0.888 (0.0963)	55,082 (19,905)	204 (14.1)	179 (23.4)	28,230 (3865)
3 (M)	50	0.956 (0.0869)	602,161 (211,386)	162 (32.8)	178 (51.9)	421,119 (171,418)

* Because of volume limitations in the mice and rats, each animal was used for one time point sampling, limiting the statistical analysis of the data. M, male; F, female.

**Table 2 ijms-24-05700-t002:** Summary of plasma 3.2 mg/kg CEND-1 pharmacokinetic parameters in pancreatic cancer patients at run-in prior to chemotherapy treatment, cycle 1 day 1 of nab-paclitaxel+gemcitabine chemotherapy treatment, and cycle 6 day 1 of nab-paclitaxel + gemcitabine chemotherapy treatment.

	N	Mean	SD	N	Mean	SD	N	Mean	SD
	Run-In			Cycle 1 Day 1			Cycle 6 Day 1		
AUC_0-t_ (h*ng/mL)	8	31,330	5760	13	28,610	5561	5	30,590	11,990
AUC_0-inf_ (h*ng/mL)	7	34,560	8782	9	30,690	5454	1	33,400	-
C_max_ (ng/mL)	8	31,210	9019	13	24,050	8546	5	25,760	10,240
T_max_ (h)	8	0.117	0.176	13	0.113	0.109	5	0.063	0.014
T_last_ (h)	8	6.725	1.096	13	5.767	1.171	5	4.897	2.178
λ_last_ (1/h)	7	0.3607	0.05081	9	0.4082	0.05423	1	0.4338	-
t_1/2_ (h)	7	1.956	0.289	9	1.725	0.230	1	1.598	-
%AUC_ext_ (%)	8	11.113	6.919	13	12.838	6.833	5	22.205	8.563
CL (ml/h/kg)	7	97.6	24	9	107.5	20.86	1	95.8	-
V_z_ (ml/kg)	7	269.2	46.5	9	262.6	28.7	1	220.9	-
C_max_/D (kg*ng/mL/m)	8	9754	2818	13	7514	2671	5	8050	3199
AUC_0-inf_/D (h*kg*ng/mL)	7	10,800	2745	9	9592	1704	1	10,440	-
AUC_0-t_/D (h*kg*ng/mL)	8	9791	1800	13	8942	1738	5	9558	3748

**Table 3 ijms-24-05700-t003:** Distribution of radioactivity in tissues of perfused female BALB/c mice following a single intravenous dose administration of [^3^H]-CEND-1 at a target dose of 5 mg/kg.

		CEND-1 µg Equivalents/g					
	Sampling Time:	0.5 h			3 h		
Tissue Type	Tissue	Female 1	Female 2	Mean	Female 3	Female 4	Mean
Alimentary	Caecum contents	BLQ	BLQ	NC	0.076	0.111	0.094
Canal	Caecum mucosa	2.17	2.03	2.10	ND	ND	NC
	Large intestine contents	ND	ND	NC	BLQ	0.136	BLQ
	Large intestine mucosa	0.936	1.33	1.13	ND	ND	NC
	Small intestine contents	2.14	0.200	1.17	ND	ND	NC
	Small intestine mucosa	1.71	0.518	1.11	ND	ND	NC
	Stomach contents	5.75	1.01	3.38	ND	BLQ	NC
	Forestomach mucosa	1.56	2.07	1.82	ND	0.110	NC
	Glandular Stomach mucosa	0.503	0.749	0.626	ND	ND	NC
CNS	Brain	ND	ND	NC	ND	ND	NC
	Choroid plexus	ND	ND	NC	ND	ND	NC
	Spinal cord	ND	ND	NC	ND	ND	NC
Connective	Bone	0.070	BLQ	BLQ	ND	ND	NC
Dermal	Skin	0.392	2.52	1.46	BLQ	BLQ	BLQ
Endocrine	Adrenal gland	0.104	0.245	0.175	ND	ND	NC
	Pituitary gland	0.131	0.224	0.178	ND	ND	NC
	Thyroid gland	0.436	0.166	0.301	ND	ND	NC
Excretory/	Liver	0.451	0.787	0.619	ND	BLQ	NC
Metabolic	Kidney: Cortex	12.2	12.7	12.5	0.236	0.196	0.216
	Kidney: Medulla	30.8	34.1	32.4	0.245	0.282	0.264
	Kidney: Whole	24.3	35.6	30.0	0.221	0.254	0.238
	Urinary bladder contents	1462	910	1186	972	780	876
	Urinary bladder wall	5.18	1.86	3.52	0.618	1.62	1.12
Exocrine	Ex-orbital lachrymal gland	0.428	0.818	0.623	ND	BLQ	NC
	Harderian gland	0.185	0.162	0.174	ND	ND	NC
	Pancreas	0.839	1.67	1.25	BLQ	0.100	BLQ
	Salivary gland	0.592	1.08	0.836	ND	ND	NC
Fatty	Fat: Brown	0.410	0.698	0.554	ND	ND	NC
	Fat: White	0.148	0.212	0.180	ND	ND	NC
Ocular	Eye: Whole	0.605	0.187	0.396	0.117	0.134	0.126
Reproductive	Ovary	0.951	1.04	0.996	ND	BLQ	NC
	Uterus	5.13	5.77	5.45	BLQ	ND	NC
Respiratory	Lung	0.290	2.47	1.38	ND	BLQ	NC
Skeletal/	Muscle: Skeletal	0.259	0.147	0.203	ND	BLQ	NC
Muscular	Myocardium	0.230	0.203	0.217	ND	BLQ	NC
Vascular/	Bone marrow	0.303	0.433	0.368	ND	BLQ	NC
Lymphatic	Lymph node	2.14	2.11	2.13	ND	ND	NC
	Spleen	0.421	0.608	0.515	BLQ	BLQ	BLQ
	Thymus	0.344	0.433	0.389	ND	ND	NC
Other	Tumour	1.43	3.62	2.53	0.078	0.115	0.097

## Data Availability

The data generated in the present study may be requested from the corresponding author.

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
