# Peer review of "Assessment of the Pharmacokinetics, Disposition, and Duration of Action of the Tumour-Targeting Peptide CEND-1"

_ijms, 2023, doi:10.3390/ijms24065700_

Round 1
Reviewer 1 Report
In this manuscript the authors describe the evaluation of the biological distribution of their lead compound (CEND-1) in a variety of settings relevant to its ultimate application in clinical trials. The work is well done, the experiments well described, and the conclusions are sound. A few small changes could help to enable easier evaluation of their findings:
*In Figure 1B could the authors indicate in the structure where 3H is found
*Could the authors provide the structure of the "RGD control peptide" described in section 2.9
*The abbreviation EB is used in section 2.9 without being defined
*Could the authors provide insight into why they didn't follow radioactivity levels in the kidney for the experiment shown in Figure 4? It seems like kidney retention of compound would be a useful parameter to characterize over an 8 h time frame
*The caption for Figure 5 says that "each symbol is a single data point" but there don't appear to be symbols corresponding to each data point
*The data presented in Figure 6 is a little bit difficult to follow. The percentages shown above the graphs in panel 6b and 6c would be easier to follow if placed in a separate table. The same is true of Figure 7
*Also related to Figure 6, the timing of the different events described is not so clear to me. When there is a rightward arrow (->) does this indicate the passage of time? Perhaps the authors could add a schematic panel showing the amount of time between each step in the treatment/image acquisition process. The same is true of Figure 7.
*In Figures 6b-c it appears the CEND-1->MRI conditions show a decrease in signal intensity relative to controls. Is this a significant finding and does it have implications for imaging applications?
Author Response
Ref 1: we thank the referee for his constructive and valuable comments.
In this manuscript the authors describe the evaluation of the biological distribution of their lead compound (CEND-1) in a variety of settings relevant to its ultimate application in clinical trials. The work is well done, the experiments well described, and the conclusions are sound. A few small changes could help to enable easier evaluation of their findings:
*In Figure 1B could the authors indicate in the structure where 3H is found
Answer: According to the suggestion of the referee, we have added the information on the localization of 3H to the legend to Figure 1.
*Could the authors provide the structure of the "RGD control peptide" described in section 2.9a
Answer: We have added the information on the control peptide in the revised version of the manuscript.
*The abbreviation EB is used in section 2.9 without being defined
Answer: We have eliminated this abbreviation from the text in the revised version of the manuscript.
*Could the authors provide insight into why they didn't follow radioactivity levels in the kidney for the experiment shown in Figure 4? It seems like kidney retention of compound would be a useful parameter to characterize over an 8 h time frame
Answer: Previous whole-body distribution and tissue penetration studies conducted with FAM-labeled CEND-1 in several types of mouse cancer models indicate a highly tumor-selective accumulation of CEND-1, and that the only normal / healthy tissues that stained positive for CEND-1 were the kidneys (moderate signal) and the bladder (strong signal), an indication of elimination via the genitourinary system into urine (Sugahara et al., Cancer Cell. 2009;16:510-20, Ye et al., Bioorg Med Chem Lett 2011;21:1146-50 ). Similar small peptides are known to be often excreted unchanged into the urine. The findings of this study support the role of kidneys as a metabolic organ and, therefore, assessing the kidney retention may not provide additional information that would be relevant for this study.
*The caption for Figure 5 says that "each symbol is a single data point" but there don't appear to be symbols corresponding to each data point
Answer: We have corrected the legend to Figure 5.
*The data presented in Figure 6 is a little bit difficult to follow. The percentages shown above the graphs in panel 6b and 6c would be easier to follow if placed in a separate table. The same is true of Figure 7
*Also related to Figure 6, the timing of the different events described is not so clear to me. When there is a rightward arrow (->) does this indicate the passage of time? Perhaps the authors could add a schematic panel showing the amount of time between each step in the treatment/image acquisition process. The same is true of Figure 7.
Answer: The reviewer is right, this part was indeed not well described in our original manuscript. We have now modified the description of the experiments illustrated in Fig. 6 and 7 and the schemes of the experimental setup. Now it should be easier to follow.
*In Figures 6b-c it appears the CEND-1->MRI conditions show a decrease in signal intensity relative to controls. Is this a significant finding and does it have implications for imaging applications?
Answer: We noticed this occasionally, but we feel that this observation is not robust yet. If this would be the case, the effect of CEND-1 on contrast agent accumulation would be even higher and thus beneficial for imaging applications.
Reviewer 2 Report
This manuscript by Järveläinen et al. describes the characterisation of CEND-1 peptide in different in different in vivo models, including mice, rats, dogs, monkeys, and human pancreatic cancer patients.
CEND-1 is the prototypic peptide for tissue penetrating peptides following the CendR rule that are able to induce tumor specific tissue penetration of drugs upon co-administration. The activity of iRGD, as described in the original Cancer Cell and Science papers by Sugahara, has been questioned and some groups failed to reproduce the results, while several groups independently reproduced part of the results. Fortunately, this did not hamper the effort to translate this into the clinics, and in 2022 encouraging results of a phase I trial were reported in Lancet Gastroenterol. Hepatol. Led by CendR Therapeutics now merged with Caladrius into Lisata Therapeutics.
Here the authors report the pharmakokinetic analysis CEND-1 in pancreatic cancer patients. Detailed analysis of radioactive CEND-1 distribution in mice revealed tumor accumulation after 3 hours in mammary carcinomas, and increased tumor penetration of Gd-GTPA was measured in and hepatic carcinoma mouse model.
The manuscript is very well written, the methods used and the results are very precisely described and the results will be of great impact in the clinical translation CEND-1.
The only minor modification I might suggest, is a more precise description of the purity, scale of synthesis, and quality controls of the peptides used in animal and human studies.
Beside this, the manuscript can be accepted for publication.
Author Response
Ref 2: We are pleased that the reviewer found our study important and that it can be published with only minor modification.
This manuscript by Järveläinen et al. describes the characterisation of CEND-1 peptide in different in different in vivo models, including mice, rats, dogs, monkeys, and human pancreatic cancer patients.
CEND-1 is the prototypic peptide for tissue penetrating peptides following the CendR rule that are able to induce tumor specific tissue penetration of drugs upon co-administration. The activity of iRGD, as described in the original Cancer Cell and Science papers by Sugahara, has been questioned and some groups failed to reproduce the results, while several groups independently reproduced part of the results. Fortunately, this did not hamper the effort to translate this into the clinics, and in 2022 encouraging results of a phase I trial were reported in Lancet Gastroenterol. Hepatol. Led by CendR Therapeutics now merged with Caladrius into Lisata Therapeutics.
Here the authors report the pharmakokinetic analysis CEND-1 in pancreatic cancer patients. Detailed analysis of radioactive CEND-1 distribution in mice revealed tumor accumulation after 3 hours in mammary carcinomas, and increased tumor penetration of Gd-GTPA was measured in and hepatic carcinoma mouse model.
The manuscript is very well written, the methods used and the results are very precisely described and the results will be of great impact in the clinical translation CEND-1.
The only minor modification I might suggest, is a more precise description of the purity, scale of synthesis, and quality controls of the peptides used in animal and human studies.
Response: We have added the available details of information on the peptides used into the revised manuscript.
Beside this, the manuscript can be accepted for publication.
Reviewer 3 Report
This paper has focused on evaluation of the tumor-targeting peptide CEND-1 (iRGD) for its pharmacokinetics, distribution, and tumor penetration pre-clinically and clinically. The pharmacokinetic properties were studied at various doses in several animal models (mice, rats, dogs and monkeys) as well as some patients with metastatic pancreatic cancer. Several tumor models (murine orthotopic 4T1 tumor, TGFa/c-myc HCC, and human HCC xenograft model) were used in the work. One [3H]-labeled analog of CEND-1 was studied for its distribution in tumor bearing mice using both autoradiography and radioactivity analysis. These results indicate a favorable in vivo pharmacokinetic profile (plasma half-life: ~25 minutes in mice and 2 hours in patients) and good tumor localization of CEND-1 after intravenous administration.
Furthermore, tumor penetrations and durations were demonstrated with Evans Blue and Gd-DTPA–enhanced MRI in HCCs. The work has justified further translation of CEND-1 into clinical applications for improving therapeutic efficacy of anticancer drugs.
Questions and Discussion:
1) [3H]-CEND-1 might not be the ideal labeling analog for revealing the pharmacokinetics and bio distribution properties of CEND-1. This is because the labeling via acylation of amino group could greatly increase the lipophilicity as well as change bio-distribution and pharmacokinetics;
2) the mechanism of action for penetration, retaining, and duration need be further elucidated. This is because it might be related with the molecular interactions between CEND-1 and the anti cancer drugs (such as hydrogen bonds and van der Waals force, etc) after coadministration, in addition of effects on tumor microenvironment and others.
3) the targeting specificity of CEND-1 also needs be further demonstrated (for example, using blocking studies and other methods).
Author Response
Ref 3: We are grateful to this reviewer for important points that helped to improve the manuscript.
This paper has focused on evaluation of the tumor-targeting peptide CEND-1 (iRGD) for its pharmacokinetics, distribution, and tumor penetration pre-clinically and clinically. The pharmacokinetic properties were studied at various doses in several animal models (mice, rats, dogs and monkeys) as well as some patients with metastatic pancreatic cancer. Several tumor models (murine orthotopic 4T1 tumor, TGFa/c-myc HCC, and human HCC xenograft model) were used in the work. One [3H]-labeled analog of CEND-1 was studied for its distribution in tumor bearing mice using both autoradiography and radioactivity analysis. These results indicate a favorable in vivo pharmacokinetic profile (plasma half-life: ~25 minutes in mice and 2 hours in patients) and good tumor localization of CEND-1 after intravenous administration.
Furthermore, tumor penetrations and durations were demonstrated with Evans Blue and Gd-DTPA–enhanced MRI in HCCs. The work has justified further translation of CEND-1 into clinical applications for improving therapeutic efficacy of anticancer drugs.
Questions and Discussion:
- [3H]-CEND-1 might not be the ideal labeling analog for revealing the pharmacokinetics and bio distribution properties of CEND-1. This is because the labeling via acylation of amino group could greatly increase the lipophilicity as well as change bio-distribution and pharmacokinetics;
Response:
This is a valid point. Radiolabelling of CEND-1 was performed by orthogonal chemistry due to ease of labelling and timescales of this approach compared to a more prolonged radiosynthesis via amino acid synthesis. Whilst this approach may affect the physiochemical properties of the peptide, there was no evidence that the PK of parent was altered and thus by inference the biodistribution – in fact, the PK of the radiolabelled peptide was totally similar to CEND-1, with e.g. similar systemic half-life. The primary aim of the QWBA and QRA study was to characterize the distribution of CEND-1, which is mediated by the RGD motif, and which we do not expect to be affected by the labelling. The NRP-1 activity might affect the intensity / quantity of the signal, but most probably not the biodistribution. A number of studies have shown that nanoparticles coupled through free amines show improved tumor targeting than the unmodified nanoparticles (Wang et al., Biomaterials 2014;35:1257-66; Wang et al., Biomaterials 2013;34:4667-79; Kim et al., Pharmaceutics 2023;15:614).
- the mechanism of action for penetration, retaining, and duration need be further elucidated. This is because it might be related with the molecular interactions between CEND-1 and the anti cancer drugs (such as hydrogen bonds and van der Waals force, etc) after coadministration, in addition of effects on tumor microenvironment and others.
Response: The mechanism of tumour penetration induced by CEND-1 has been at least in part elucidated. After binding of CEND-1 to tumour-specific integrins at the tumor vascular endothelium, CEND-1 is cleaved by a protease and a fragment of CEND-1 is able to bind to neuropilin-1. This causes tumor extravasation of the peptide and of co-administered payloads by a transendothelial endocytic mechanism that resembles VEGF-A-induced transport vesicles called vesiculo-vacuolar organelles (VVOs) (Liu et al., J. Clin. Invest. 2017, 127, 2007-2018). CEND-1/iRGD then activates a transcellular transport pathway in the extravascular tumor tissue, effectively transforming the solid tumour microenvironment into a temporary drug conduit that facilitates penetration of chemotherapeutics co-administered with CEND-1 into tumours. The vesicular transport was found to be active in tumor vessels of pancreatic cancer at 24 h after injection of CEND-1, although the peptide is eliminated from the blood and tumour tissue. The mechanisms mediating the prolonged duration of the tumor-permeabilising effect is currently unknown and important to elucidate, but is not easy to fulfil and was beyond the scope of this study. This also holds for the elucidation of the effect of CEND-1 on the tumor microenvironment.
CEND-1 can be coupled covently to payloads and can be used to achieve deep tumour penetration of anti-cancer drugs. However, there is no evidence that an interaction between payloads and CEND-1/iRGD is involved in tumour penetration, when the two substances are co-administered in tumour mice. As the peptide has only a short plasma half live (Pang et al., J. Control. Release, 2014, 175, 48-53), we believe that at least the prolonged stimulation of the CEND-1-triggered transport of the peptide and co-administered payloads does not involve interaction of CEND-1 with payloads.
3) the targeting specificity of CEND-1 also needs be further demonstrated (for example, using blocking studies and other methods).
Response: The targeting specificity of CEND-1/iRGD has been assessed in a high number of publications. These studies together show that CEND-1 has a high specificity for tumours. This has been recently reviewed by E. Ruoslahti (Proc Natl Acad Sci U S A 2022;119:e220018311).